# Where are we in the search for an Artificial Visual Cortex for Embodied Intelligence?

Arjun Majumdar[1][*] Karmesh Yadav[2][*] Sergio Arnaud[2][*]
Yecheng Jason Ma[23] , Claire Chen[4] Sneha Silwal[2] Aryan Jain[5]
Vincent-Pierre Berges[2] Pieter Abbeel[2] Jitendra Malik[5,2] Dhruv Batra[12]
Yixin Lin[2][†] Oleksandr Maksymets[2][†] Aravind Rajeswaran[2][†] Franziska Meier[2][†]

## Abstract

We present the largest and most comprehensive empirical study of pre-trained visual representations (PVRs) or visual 'foundation models' for Embodied AI. First, we curate CortexBench, consisting of 17 different tasks spanning locomotion, navigation, dexterous, and mobile manipulation. Next, we systematically evaluate existing PVRs and find that none are universally dominant.

To study the effect of pre-training data scale and diversity, we combine over 4,000 hours of egocentric videos from 7 different sources (over 5.6M images) and ImageNet to train different-sized vision transformers using Masked Auto-Encoding (MAE) on slices of this data. Contrary to inferences from prior work, we find that scaling dataset size and diversity does *not* improve performance universally (but does so on average).

Our largest model, named **VC-1**, outperforms all prior PVRs on average but does not universally dominate either. Finally, we show that task- or domain-specific adaptation of **VC-1** leads to substantial gains, with **VC-1** (adapted) achieving competitive or superior performance than the best known results on all of the benchmarks in CortexBench. These models required over 10,000 GPU-hours to train and can be found on our website for the benefit of the research community.

## 1 Introduction

Eyesight is considered one of the greatest inventions of biological evolution (Lane, 2010). Out of the 38 known phyla in the animal kingdom, only 6 have evolved eyes yet they account for 95% of all species (Lane, 2010) – vision seems to confer an enormous advantage. Of course, the evolution of visual *sensing* via eyes progresses in concordance with visual *perception* – via a visual cortex, the region of the brain that (together with the motor cortex) enables an organism to convert sight into movement. In this work, we ask the same question Fukushima (Fukushima, 1975; 1980) did nearly 50 years ago – how do we design an *artificial visual cortex*, the module in a larger computational system that enables an artificial agent to convert camera input into actions?

In contemporary AI, this question has been operationalized as the design of pre-trained visual representations (PVRs) or visual 'foundation models' for embodied AI (EAI).[1] Indeed, recent work has shown that PVRs trained on large quantities of egocentric-videos and web-images can substantially improve performance and learning efficiency for navigation (Khandelwal et al., 2022; Yadav et al., 2022b; 2023) and manipulation tasks (Parisi et al., 2022; Nair et al., 2022; Radosavovic et al., 2022; Ma et al., 2022). Unfortunately, each study is fundamentally incommensurable, as each uses different self-supervised learning (SSL) algorithms on different pre-training datasets, designed for, and evaluated on different downstream EAI tasks.

---

[*]Equal contribution † Equal contribution [1]Georgia Institute of Technology [2]Meta AI [3]University of Pennsylvania [4]Stanford University [5]UC Berkeley. Correspondence to: <cortex@meta.com>

[1]We use embodied AI (EAI) as an umbrella term for all communities studying visuomotor control such as robot learning, vision-based reinforcement learning, egocentric computer vision, etc.

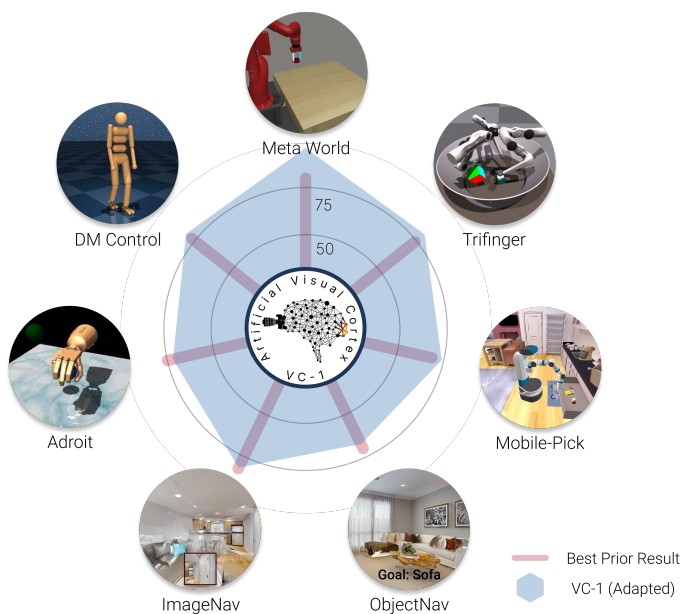

Figure 1: An artificial visual cortex for embodied intelligence must support a diverse range of sensorimotor skills, environments, and embodiments; we curate CORTEXBENCH to systematically measure progress towards this ambitious goal. Our strongest model, denoted **VC-1** (adapted) above, is competitive with or outperforms the *best prior results* (success rates) on all benchmarks in CORTEXBENCH. Notice that this comparison is particularly unforgiving because the best prior results are benchmark-specific and not constrained to share any aspect of their design.

Naturally, one might ask: is there a universally-dominant configuration? Essentially, *does an artificial visual cortex already exist?*[2]

To answer this question, we conduct the largest and most comprehensive empirical study to-date of visual foundation models for EAI. First, we curate CORTEXBENCH, a new benchmark for evaluating PVRs, consisting of 17 tasks spanning low-level locomotion (Tassa et al., 2018), table-top manipulation of rigid and articulated objects Yu et al. (2020), dexterous manipulation (Rajeswaran et al., 2018), multi-finger coordinated manipulation (Wüthrich et al., 2020), indoor visual navigation (Savva et al., 2019a), and mobile manipulation (Szot et al., 2021). The visual environments span from flat infinite planes to table-top settings to photorealistic 3D scans of real-world indoor spaces. The agent embodiments vary from stationary arms to dexterous hands to idealized cylindrical navigation agents to articulated mobile manipulators. The learning conditions vary from few-shot imitation learning to large-scale reinforcement learning. The exhaustiveness of this study enables us to draw conclusions with unprecedented scope and confidence.

Our first finding is a *negative result*. We discover that while existing PVRs generally outperform learning-from-scratch baselines, none is universally dominant. Instead, we find that PVRs tend to work best in the domains (locomotion, manipulation, navigation) they were originally designed for. We note that no claims of universality were made in prior work, so this finding is illustrative rather than refutative. Overall, serendipity did not come to pass – an artificial visual cortex does not already exist.[3] However, curiously, the *kinds of PVRs* that are locally-dominant in CORTEXBENCH differ significantly in the size and type of pre-training datasets: CLIP Radford et al. (2021) was pre-trained on 400M image-text pairs from the web; MVP Radosavovic et al. (2022) on 4.5M frames from web-images and many egocentric-video datasets; R3M Nair et al. (2022) on ∼5M frames from Ego4D – yet, each performs best on some subset of tasks in CORTEXBENCH. This leads to a natural question: *how does scaling model size, dataset size, or diversity affect performance on CORTEXBENCH?*

---

[2]To the degree of our ability to measure it.

Can we use scaling as a means to learn a single PVR that works for all of the diverse tasks in CortexBench?

To study these questions, we combine over 4,000 hours of egocentric videos from 7 different sources containing humans manipulating objects and navigating indoor spaces encountered in daily life, together with ImageNet. From this union, we create 4 pre-training datasets of varying size and diversity, with the largest containing over 5.6M images. We train vision transformers (ViT-B and ViT-L) Dosovitskiy et al. (2020) on these 4 datasets using Masked Auto-Encoding (MAE) He et al. (2021), and systematically analyze their performance on CortexBench. To benefit the EAI community, we will open-source these models, which required over 10,000 GPU hours to train.

We do find evidence supporting the scaling hypothesis, but the picture that emerges is more nuanced than what a superficial reading might suggest. Our largest model trained on all data, named **VC-1**, outperforms the best existing PVR by 1.2% on average. However, **VC-1** does *not* universally dominate either – i.e., there are PVRs trained on smaller amounts of data that outperform it on specific tasks. A similar trend emerges for data diversity – more is better on average, but not universally. For instance, the best performance on the `Mobile-Pick` task from Habitat 2.0 Szot et al. (2021) is achieved by pre-training on the subset of video data focused on manipulation; presumably because the mobility involved in the task is fairly limited. Thus, our second key finding is: *Naively scaling dataset size and diversity does not improve performance uniformly across benchmarks.*

Our findings reveal a challenge and opportunity for the community – the search for a PVR that is universally dominant (or 'foundational') for EAI calls for innovations in architecture, learning paradigm, data engineering, and more. As the final step in this paper, but as a first step towards this open problem, we study *adapting* **VC-1** with either task-specific training losses or datasets (via MAE He et al. (2021)) to specialize **VC-1** for each domain. We find that adapting **VC-1** results in it becoming competitive with or outperforming the *best prior results on all of the benchmarks* in CortexBench. We highlight that this comparison is particularly unforgiving, since best prior results are highly domain-specific and are not constrained to share any aspect of their design. To our knowledge, **VC-1** (adapted) is the first PVR that is competitive with (or outperforms) state-of-art results on such a diverse set of EAI tasks ( Figure 1).

We will release code for CortexBench to enable the EAI, robotics, and CV communities to benchmark their own models, and share our pre-trained models (including **VC-1**) that we believe can serve as a starting point for all visuomotor tasks of interest today.

## 2 Related Work

**Pre-trained visual representations (PVRs).** The last few years have seen increasing interest in the self-supervised learning (SSL) of visual representations He et al. (2021); Caron et al. (2020); Baevski et al. (2022b); Chen et al. (2020; 2021). These algorithms use contrastive Chen et al. (2020; 2021), distillation-based Caron et al. (2020); Baevski et al. (2022b), or reconstructive Bao et al. (2021); He et al. (2021) objectives for training. Recently, a flurry of works have proposed using the vision transformers (ViTs) Dosovitskiy et al. (2021) with masked image modeling He et al. (2021); Baevski et al. (2022a); Yao et al. (2022), which among other benefits reduces the computation time required for pre-training. In this work, we use one such pre-training algorithm (MAE He et al. (2021)) to explore scaling and adapting pre-trained visual representations (PVRs).

**PVRs for embodied AI.** Inspired by the advancements in self-supervised learning, recent work has incorporated visual representation learning into the training pipelines for EAI agents (Parisi et al., 2022; Nair et al., 2022; Radosavovic et al., 2022; Ma et al., 2022; Khandelwal et al., 2022; Yadav et al., 2022b; 2023). Specifically, Parisi et al. (2022) evaluate several PVRs trained with supervised or self-supervised learning on a range of EAI tasks, demonstrating promising results under a few-shot imitation learning evaluation protocol. Nair et al. (2022); Radosavovic et al. (2022); Ma et al. (2022) introduce new methods for pre-training visual representations using egocentric video data, targeting robotic manipulation tasks. Similarly, Khandelwal et al. (2022); Yadav et al. (2022b; 2023) use pre-trained visual representations

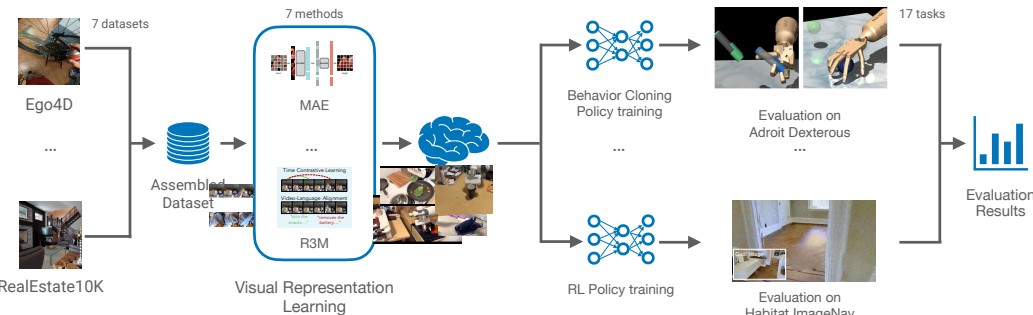

Figure 2: Overview of CORTEXBENCH. We assemble relevant datasets and visual representation learning algorithms to produce candidate Visual Cortex models, which are then evaluated using either reinforcement or imitation learning on a set of highly diverse tasks.

to improve performance on multiple visual navigation tasks. Closely related, Radosavovic et al. (2022) demonstrate that MAE pre-training on internet-scale video and image data can produce effective visual representations for robotic manipulation tasks. In contrast, our work studies a larger range of embodied AI tasks (collected in CORTEXBENCH) to understand how PVRs can provide a general-purpose foundation for embodied agents and explores in-domain model adaptation for various tasks.

**Scaling model and dataset size.** Several works have showed that scaling model and dataset size improves performance on vision tasks like image classification (Zhai et al., 2022; Tian et al., 2021; Goyal et al., 2021). In EAI, Radosavovic et al. (2022) find that scaling model and data sizes improves downstream policy performances for robotic manipulation tasks. While such prior works have been confined to narrow domains like image classification and robotic manipulation, our work is the first to study if scaling can provide better models on a broad range of EAI tasks.

**Adapting PVRs.** When and how to adapt PVRs for downstream applications remains an open research question (Kumar et al., 2022; Wijmans et al., 2022; Kirichenko et al., 2022; Lee et al., 2022; Goyal et al., 2022). In the context of EAI, Parisi et al. (2022) and Hansen et al. (2022b) show that naively fine-tuning PVRs with behavior cloning can reduce performance in simulation, and Radosavovic et al. (2022) observe minimal gains in real-world tasks manipulation tasks. In large-scale RL settings, Yadav et al. (2022b; 2023) show that end-to-end finetuning considerably improves performance for indoor visual navigation. By comparison, Pari et al. (2021) find simple $k$-nearest-neighbor adaptation works well for real-world visual imitation tasks. Our work neither aims nor expects to be the final word on this fertile topic.

## 3 BENCHMARKING PROGRESS TOWARDS AN ARTIFICIAL VISUAL CORTEX FOR EMBODIED AI

This section describes CORTEXBENCH, a curated set of EAI tasks designed to evaluate the ability of pre-trained visual representations (PVRs) to support a wide variety of EAI applications. Specifically, CORTEXBENCH includes 17 tasks drawn from 7 existing EAI benchmarks as shown in Figure 1 and described in Appendix B. For each task, we delineate a downstream policy learning paradigm (e.g., few-shot imitation learning) and evaluation protocol that follows community standards in each domain (detailed in Appendix C). By fixing the tasks and downstream learning methods as shown in Figure 2, we are able to focus our evaluations on the contribution of PVRs, which allows us to measure progress towards the development of an artificial visual cortex for embodied intelligence. We use CORTEXBENCH to conduct the largest and most comprehensive empirical study to-date of PVRs from prior work (Section 3.1).

We recommend two metrics to evaluate PVR performance: **Mean Success** and **Mean Rank**. **Mean Success**: the average success rate across all benchmarks. **Mean Rank**: for each benchmark, we rank PVRs based on their success rate; then we average these rankings across all benchmarks.

Table 1: Performance of different **frozen** pre-trained visual representations on a diverse suite of evaluation domains. Best prior results means that the results are the best reported in literature prior to this work. Overall, we find that no single PVR consistently performs the best across all benchmarks. However, we find that several of these pre-trained models often outperform a random training from scratch baseline. Best prior results sources (row 1): Adroit and MetaWorld approximated from Nair et al. (2022), DMControl from Parisi et al. (2022), ImageNav from Yadav et al. (2022b), ObjectNav from Ramrakhya et al. (2023). Frozen PVR Sources (row 2): Adroit, MetaWorld, and DMControl are the same as SOTA, ImageNav from Yadav et al. (2022b), ObjectNav from Deitke et al. (2022b).

| # | Model | Imitation Learning | | | | | Reinforcement Learning | | Mean | |
|---|---|---|---|---|---|---|---|---|---|---|
| | | Adroit | MetaWorld | DMControl | Tri-Finger | ObjectNav | ImageNav | Mobile Pick | Rank | Success |
| 1 | Best prior result (any setting) | 75 | 80 | 77 | - | 70.4 | 82.0 | - | | |
| 2 | Best prior result (Frozen PVR) | 75 | 80 | 77 | - | 54.4 | 61.8 | - | | |
| 3 | Random (ViT-B) Frozen | 2.0 ± 2.0 | 0.5 ± 0.5 | 10.1 ± 0.6 | 57.8 ± 0.5 | 19.2 ± 0.9 | 42.1 ± 0.8 | 10.8 ± 1.4 | 7.2 | 20.4 |
| 4 | Random (ViT-L) Frozen | 2.7 ± 1.8 | 0.5 ± 0.5 | 9.1 ± 0.2 | 57.2 ± 0.9 | 19.3 ± 0.9 | 45.2 ± 0.8 | 20.6 ± 1.8 | 6.9 | 22.1 |
| 5 | Random (ViT-B) Fine-tuned | 44.0 ± 2.0 | 49.9 ± 7.3 | 43.5 ± 2.4 | 56.1 ± 1.3 | 28.5 ± 1.0 | 62.5 ± 0.7 | 47.6 ± 2.2 | 5.3 | 47.4 |
| 6 | MVP (ViT-B) | 48.0 ± 3.3 | 91.2 ± 2.9 | 65.9 ± 2.4 | 59.7 ± 0.3 | 51.2 ± 1.1 | 64.7 ± 0.7 | 56.0 ± 2.2 | 3.1 | 62.4 |
| 7 | MVP (ViT-L) | 53.3 ± 4.1 | 87.5 ± 3.4 | 69.2 ± 1.5 | 74.1 ± 0.3 | 55.0 ± 1.1 | 68.1 ± 0.7 | 65.4 ± 2.1 | 2.1 | 67.5 |
| 8 | CLIP (ViT-B) | 47.3 ± 3.0 | 75.5 ± 3.4 | 55.5 ± 1.4 | 62.0 ± 0.5 | 56.6 ± 1.1 | 52.2 ± 0.8 | 49.8 ± 2.2 | 3.9 | 57.0 |
| 9 | VIP (RN-50) | 54.0 ± 4.8 | 90.1 ± 2.2 | 72.5 ± 2.7 | 66.7 ± 0.2 | 26.4 ± 1.0 | 48.8 ± 0.8 | 7.2 ± 1.2 | 4.0 | 52.3 |
| 10 | R3M (RN-50) | 73.3 ± 2.0 | 96.0 ± 1.1 | 81.1 ± 0.7 | 69.2 ± 0.8 | 22.7 ± 0.9 | 30.6 ± 0.7 | 33.2 ± 2.1 | 3.4 | 58.0 |

## 3.1 Do we already have a foundation model?

First, we evaluate several existing pre-trained visual representations (PVRs) on COR-TEXBENCH to study whether existing open-sourced visual backbones can consistently perform well across all tasks. For all evaluations we consider frozen visual representations to disentangle the effect of learned representations from downstream task learning. Specifically, we include the following models:

– CLIP (Radford et al., 2021) Contrastive image-language pre-training objective; Trains on 400M images-text pairs from the internet (WIT); ViT-B backbone.
– R3M (Nair et al., 2022) Time-Contrastive video-language alignment pre-training objective; Trains on 5M images from a subset of Ego4D; ResNet-50 backbone.
– MVP (Radosavovic et al., 2022). Pre-trains with MAE; Trains on 4.5M images from Egocentric videos and ImageNet; ViT-B and ViT-L backbones.
– VIP (Ma et al., 2022). Goal-conditioned value function pre-training objective; Trains on 5M images from a subset of Ego4D; ResNet-50 backbone.

These models cover a wide range of architectures, pre-training objectives, and pre-training datasets, constituting a solid set for comparisons. Additionally, we include randomly initialized ViTs with both frozen weights and fine-tuned weights to assess the necessity of pre-training and the limitations of pure end-to-end in-domain learning.

Table 1 shows the evaluation results aggregated by benchmark; no single model excels in all cases. Among all of the models evaluated, R3M performs the best on Adroit, MetaWorld, and DMControl. While MVP (ViT-L) performs best on Trifinger, ImageNav, and Mobile Pick. CLIP, on the other hand, achieves the best results on ObjectNav.

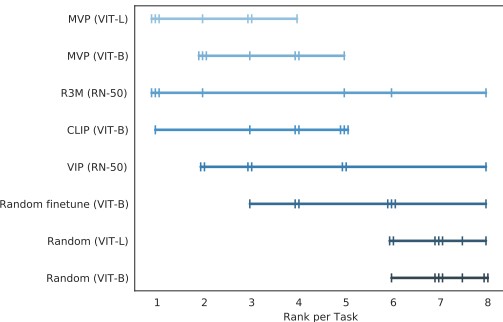

Figure 3: Rank distribution per model. For every model, we compute the ranks it achieved on each of the 7 benchmarks. We visualize them as vertical lines, where each rank number $x$ receives a tick if that model achieved such rank $x$. For instance, MVP (ViT-L) achieves ranks 1,1,1,2,3,3,4 across the 7 benchmarks. Significant variability exists in the performance of PVRs across benchmarks.

The variance in performance of existing PVRs on CORTEXBENCH is further illustrated in Figure 3. Indeed, PVRs can be successful on some benchmarks but fail on others; for instance, while CLIP is the best model for ObjectNav (ranking first), its performance is poor on Adroit and MetaWorld (ranking fifth). This variance highlights that we do not yet have one strong performing artificial visual cortex for embodied AI yet.

# 4 ANALYZING THE SCALING HYPOTHESIS FOR EAI

The previous section investigated models pre-trained on datasets of varying size and diversity. Interestingly, while the model pre-trained on the largest dataset (CLIP) performs well on one benchmark (ObjectNav) it does not perform well across all tasks. We now ask: how much does the relevance and diversity of the pre-training dataset and the model size matter? To study this, we fix the pre-training objective – MAE (He et al., 2021) – and then vary the composition of the pre-training dataset and the size of the visual backbone (ViT-B with 86M parameters and ViT-L with 307M parameters). We measure the corresponding changes in performance on CORTEXBENCH. MAE is selected for these experiments due to the strong performance on CORTEXBENCH of the MVP Radosavovic et al. (2022) models (Table 1), which use the MAE pre-training objective.

## 4.1 CONSTRUCTING A PRE-TRAINING DATASET FOR EAI

Table 2: Datasets assembled to study effects of pre-training dataset size, diversity, and relevance – the largest of which (**Ego4D+MNI**) has 5.6M frames. For a detailed breakdown of the composition of each dataset, see Table 6 in Appendix D

| Name | Frames Used |
| --- | --- |
| **Ego4D** | 2,790,520 |
| **Ego4D+M** (Manipulation) | 3,538,291 |
| **Ego4D+N** (Navigation) | 3,593,049 |
| **Ego4D+MN** (Manipulation, Navigation) | 4,340,820 |
| **Ego4D+MNI** (Manipulation, Navigation, ImageNet) | 5,621,987 |

To evaluate the impact of dataset size and diversity on our benchmark tasks, which involve various navigation and manipulation challenges, we employ a combination of nine datasets. These datasets include Ego4D Grauman et al. (2022), 100 Days of Hands (100DOH) Shan et al. (2020), Something-Something v2 (SS-V2) Goyal et al. (2017), and Epic Kitchens Damen et al. (2018). This subset consists of videos showcasing people manipulating objects and are comparable to the datasets used in MVP Radosavovic et al. (2022). Additionally, we use two egocentric indoor navigation datasets: the Real Estate 10K dataset Zhou et al. (2018) and the OpenHouse24 dataset (described in Appendix D.1). Finally, we include ImageNet Deng et al. (2009) as a representative static internet image dataset.

We strategically select combinations of these datasets (listed in Table 2 and below) to answer the following questions:

– What is the impact of scaling dataset size and diversity?
– How does the inclusion of *less-relevant* datasets influence the performance of PVRs on embodied AI tasks?

**Ego4D** Grauman et al. (2022) is our base pre-training dataset and encompasses a wide range of egocentric videos consisting of *daily life activities* such as home, leisure, transportation, and workplace activities.

**Ego4D+M** extends **Ego4D** with three object manipulation-centric datasets: 100DOH, SS-v2, and Epic Kitchens. This results in a dataset comprising 3.5 million frames that is primarily focused on manipulation scenarios.

**Ego4D+N** extends **Ego4D** with two egocentric indoor navigation datasets: OpenHouse24 and RealEstate10K. This results in a dataset with 3.5 million frames, which is similar in size to **Ego4D+M**, but is more diverse because it contains a larger proportion of navigation data than the manipulation-centric datasets **Ego4D** and **Ego4D+M**[3].

---

[3]While **Ego4D** does contain navigation data (e.g., people moving from location to another), the dataset is heavily skewed towards object manipulation activities.

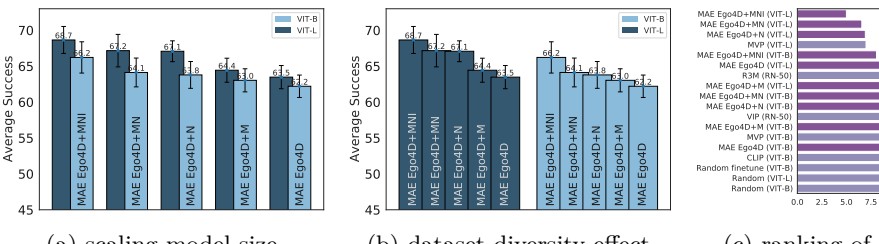

(a) scaling model size     (b) dataset diversity effect     (c) ranking of all models

Figure 4: Scaling experiments: Visualizing model performance averaged across all benchmarks in Table 3. Overall, we demonstrate modest but positive scaling trends in both (a) scaling model size, and (b) dataset diversity. c) Average ranking across all benchmarks. We compare existing PVRs (baselines) (Table 1) and scaling models (Table 3) by showcasing their ranking across all benchmarks, **VC-1**: **Ego4D+MNI** (ViT-L) achieves the highest average rank.

**Ego4D+MN** combines **Ego4D** with both the three object manipulation-centric datasets and two indoor navigation dataset, resulting a dataset with 4.3 million frames. While larger than **Ego4D+M** and **Ego4D+N**, it does not include any new types of data beyond the manipulation and navigation videos in the previous subsets. Thus, it is no more diverse than **Ego4D+N** (which includes both types of data).

**Ego4D+MNI** includes **Ego4D**, all of the manipulation-centric and indoor navigation datasets, and ImageNet for a total of 5.6M frames. This dataset allows us to explore the impact of static internet images on our benchmark tasks.

## 4.2 Scaling Hypothesis Findings

We now turn to analyzing the effect of increasing model size, dataset size, and dataset diversity. The full set of results is shown in Figure 4 and Table 3. The key takeaways are:

**Model Size.** We find that increasing model size positively impacts performance on Cortex­Bench. Specifically, in Figure 4a, we find that with all pre-training datasets, switching from ViT-B to ViT-L improves average performance on CortexBench. However, in Table 3, we find exceptions where this general trend does not hold. For instance, when pre-trained on **Ego4D+MNI**, the ViT-B model outperforms the ViT-L model on MetaWorld and Trifinger.

**Dataset Size and Diversity.** Figure 4b shows that, in general, increasing dataset size and diversity improves performance. Models are are ordered from right-to-left by increasing size and the diversity of their pre-training dataset. We mostly see improvements for ViT-B and ViT-L.

For instance, **Ego4D+M** slightly improves upon **Ego4D** by 0.6 and 0.9 points (62.2 → 62.8 and 63.5 → 64.4) in the case of ViT-B and ViT-L, respectively. The gains with **Ego4D+N** are larger and it outperforms **Ego4D** by 1.6 points using ViT-B (62.2 → 63.8) and by 3.6 points for ViT-L (63.5 → 67.1). We find that **Ego4D+N** has a larger improvement over the base **Ego4D** dataset than **Ego4D+M**, even though **Ego4D+N** and **Ego4D+M** dataset are similar in size. These results demonstrate that increasing diversity by adding indoor navigation data improves performance more than adding additional manipulation data to **Ego4D**.

Additionally, we find that pre-training on **Ego4D+MN** is roughly on par with pre-training on **Ego4D+N**. We see a 0.3 and 0.1 point difference (63.8 → 64.1 and 67.1 → 67.2) for ViT-B and ViT-L, respectively, even though **Ego4D+MN** has about 800K more training frames. Together with the results above, this demonstrates that increasing data diversity seems to matter more than simply increasing dataset size.

Next, we analyze the effect of including static internet image data. Specifically, we find that adding ImageNet positively impacts average performance on CortexBench. For example, models pre-trained on **Ego4D+MNI** outperform those pre-trained on **Ego4D+MN** by 1.9 points (64.1 → 66.2) for ViT-B and 1.5 points (67.2 → 68.7) for ViT-L. This finding further highlights the importance of seeking data diversity to build better representations.

Finally, on average, our largest model (ViT-L) pre-trained on all datasets (**Ego4D+MNI**), achieves the highest rank when averaged across all benchmark tasks (Table 3 row 11), with

Table 3: Average success per benchmark of scaling hypothesis models. We highlight the best model from the set of models trained to evaluate the scaling hypothesis in bold. We find that on average the **VC-1 Ego4D+MNI** (VIT-L) model performs best, but is not the best for each benchmark. Our best model outperforms in-domain from scratch learning on all benchmarks.

| # | Benchmark | Adroit | Meta-World | DMControl | Trifinger | ObjectNav | ImageNav | Mobile Pick | Mean Rank | Mean Success |
|---|-----------|--------|-----------|-----------|-----------|-----------|----------|-------------|-----------|--------------|
| 1 | Best prior result (any setting) | 75 | 80 | 77 | - | 70.4 | 82.0 | - | | |
| 2 | Rand (ViT-B) fine-tuned | 44.0 | 49.9 | 34.2 | 55.0 | 28.5 | 65.0 | 47.6 | | |
| 3 | Best result Table 1 (Frozen PVR) | 73.3 | 96.0 | 81.1 | 74.1 | 56.6 | 68.1 | 65.4 | | |
| 4 | Ego4D (VIT-B) | 48.7 ± 1.3 | 86.1 ± 2.1 | 64.1 ± 2.3 | 68.3 ± 1.1 | 46.8 ± 1.1 | 64.0 ± 0.7 | 57.4 ± 2.2 | 8.6 | 62.2 |
| 5 | Ego4D (VIT-L) | 50.0 ± 1.2 | 92.9 ± 2.4 | 60.8 ± 3.3 | 69.7 ± 0.5 | 47.6 ± 1.1 | 55.8 ± 0.8 | 67.6 ± 2.1 | 5.9 | 63.5 |
| 6 | Ego4D+N (VIT-B) | 50.0 ± 2.4 | 86.4 ± 2.9 | 59.5 ± 2.4 | 67.8 ± 1.3 | 54.7 ± 1.1 | 68.7 ± 0.7 | 59.4 ± 2.2 | 7.2 | 63.8 |
| 7 | Ego4D+N (VIT-L) | 54.0 ± 1.2 | 89.1 ± 2.9 | 66.4 ± 1.7 | 66.9 ± 0.4 | 57.4 ± 1.1 | 70.5 ± 0.7 | 65.2 ± 2.1 | 3.5 | 67.1 |
| 8 | Ego4D+M (VIT-B) | 51.3 ± 2.4 | 83.5 ± 2.6 | 64.3 ± 1.8 | 69.1 ± 0.4 | 47.3 ± 1.1 | 65.8 ± 0.7 | 59.8 ± 2.2 | 7.0 | 63.0 |
| 9 | Ego4D+M (VIT-L) | 52.0 ± 1.3 | 88.3 ± 3.2 | 64.7 ± 2.4 | 64.7 ± 0.9 | 47.3 ± 1.1 | 65.5 ± 0.7 | 68.6 ± 2.1 | 6.0 | 64.4 |
| 10 | Ego4D+MN (VIT-B) | 48.7 ± 2.4 | 85.3 ± 5.2 | 64.2 ± 1.9 | 70.3 ± 0.5 | 52.8 ± 1.1 | 68.9 ± 0.7 | 58.6 ± 2.2 | 6.9 | 64.1 |
| 11 | Ego4D+MN (VIT-L) | 52.7 ± 4.2 | 86.7 ± 3.9 | 69.7 ± 3.3 | 72.4 ± 0.5 | 58.4 ± 1.1 | 69.1 ± 0.7 | 61.2 ± 2.2 | 3.1 | 67.2 |
| 12 | Ego4D+MNI (VIT-B) | 54.0 ± 4.0 | 89.6 ± 3.9 | 63.8 ± 2.7 | 72.2 ± 0.6 | 55.4 ± 1.1 | 67.9 ± 0.7 | 60.6 ± 2.2 | 4.4 | 66.2 |
| 11 | **VC-1**: Ego4D + MNI (VIT-L) | 59.3 ± 5.2 | 88.8 ± 2.2 | 66.9 ± 1.4 | 71.7 ± 0.4 | 60.3 ± 1.1 | 70.3 ± 0.7 | 63.2 ± 2.2 | 2.4 | 68.7 |

a mean rank of 2.4. This performance is superior to the second-best model (**Ego4D+MN** ViT-L, Table 3 row 9) that has an average rank of 3.1. We call this model **VC-1**.

However, upon further dis-aggregation, we find that while **VC-1** performs best on average, it is not the best for each benchmark. For example, the best model for Mobile Pick, a mobile manipulation task, is a ViT-L trained on **Ego4D+M** and the best model for ImageNav, an indoor navigation task, is the ViT-L trained on **Ego4D+N**. These findings suggest that task-specific pre-training datasets could enhance the performance of models on individual tasks. However, it is important to note that this approach would lead to multiple pre-trained models, each tailored to a specific task, and not a unified visual foundation model.

### 4.3 How does **VC-1** compare to existing PVRs?

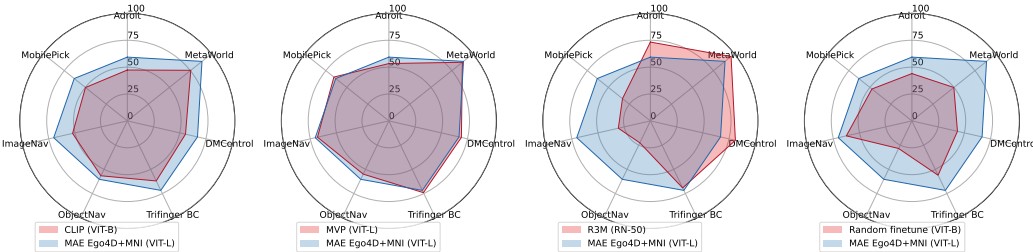

Figure 5: Comparison of **VC-1** with existing PVRs. **VC-1** matches or exceeds existing PVRs on all benchmarks except R3M on AD, MW, and DMC, indicating an opportunity for model adaptation. We now compare **VC-1** with existing PVRs from Section 3.1. On average it ranks as the best model across all benchmarks Figure 4c. We focus on R3M, MVP, and CLIP, which achieved the highest success in at least one benchmark; and compare to fine-tuning from scratch. In terms of mean success, **VC-1** (Table 3 row 11) outperforms MVP (ViT-L) by +1.2 points (67.5 → 68.7), R3M by +10.7 (58.0 → 68.7), CLIP by +11.7 (57.0 → 68.7), and end-to-end fine-tuning from scratch +19.6 (49.1 → 68.7).

Impressively, **VC-1** outperforms CLIP *on every benchmark* (Figure 5), despite training on a 70X smaller dataset, emphasizing the importance of egocentric interaction datasets. **VC-1** also outperforms fine-tuning from scratch on every benchmark, indicating that PVRs trained with out-of-domain data can outperform end-to-end learning.

When compared to R3M, **VC-1** demonstrates superior performance on average and on 4 out of 7 benchmarks (Figure 5). It is outperformed by R3M on Adroit, MetaWorld and DMControl benchmarks. It is unclear whether this gap is caused by the different training objective, pre-training dataset, or backbone. This highlights the need for comparable evaluations on benchmarks like CORTEXBENCH.

The MVP model is the most similar in terms of results, architecture, and pre-training objective to **VC-1**, with the main difference being the addition of a *convolutional stem* in MVP. **VC-1** outperforms MVP VIT-L by 1.3 points on mean success and performs better on four out of seven benchmarks, likely due to the use of a more diverse dataset.

Overall, **VC-1** is an effective model across a broad set of tasks and thus a reasonable starting point for novel EAI problems. However, it is not always the best performing model for a specific task. This leads us to theorize that there is a domain gap that might be bridged with dataset engineering or adaptation of the PVR.

## 5 ADAPTING **VC-1**

Table 4: Adapting **VC-1** with end-to-end fine-tuning or self-supervised learning (MAE) on in-domain data leads to substantial gains.

| # | Method | Adroit | MetaWorld | DMControl | Tri-Finger | ObjectNav | ImageNav | Mobile Pick |
|---|--------|--------|-----------|-----------|------------|-----------|----------|-------------|
| 1 | Best prior result (any setting) | 75 | 80 | 77 | - | 70.4 | 82.0 | - |
| 2 | Best result from our experiments | 73.3 | 96.0 | 81.1 | 74.1 | 60.3 | 70.5 | 68.6 |
| 3 | In-domain MAE baseline | 47.3 | 83.4 | 77.6 | 80.4 | 39.9 | 47.6 | 51.6 |
| 4 | **VC-1** | 59.3 | 88.8 | 66.9 | 71.7 | 60.3 | 70.3 | 63.2 |
| 5 | **VC-1** E2E fine-tuning | 15.9 | 22.7 | 6.7 | 70.9 | 67.7 | 81.6 | 74.0 |
| 6 | **VC-1** MAE adaptation | 72.0 | 96.0 | 80.9 | 80.6 | 57.4 | 67.0 | 62.4 |

In prior sections, we focused on evaluating **VC-1** as a **frozen** PVR for EAI. We now study if *adapting* **VC-1** can improve results in downstream tasks. We use a broad definition of adaptation (Bommasani et al., 2021), which, in the context of large pre-trained foundation models, can take several forms from simple prompting (Wei et al., 2022), to selectively updating some or all weights of the backbone (Kumar et al., 2022; Hansen et al., 2022a; Yadav et al., 2023).

In the context of PVRs for EAI, adaptation can serve at least two purposes. The first is **task-specialization** in the feature extraction stage. Since **VC-1** was trained with MAE He et al. (2021), it captures features that are generally useful for reconstructing images. Adaptation can specialize the visual backbone to extract features required for performing specific downstream tasks such as object rearrangement. Secondly, adaptation can also help **mitigate domain-gap** that might exist between pre-training and evaluation settings. In general, domain-gap can arise for several reasons such as poor coverage in pre-training data collection or deployment in novel conditions (e.g., on robots) not seen in the pre-training data (e.g., in human-centric video datasets). Domain gap is naturally instantiated in our setup, since **VC-1** was pre-trained on real-world, human video data while our downstream evaluation in CORTEXBENCH uses simulated EAI domains with different visual characteristics.

**End-to-end (E2E) fine-tuning** with a task-specific loss function can in-principle capture both of the aforementioned benefits of adaptation, and is widely used in computer vision literature (He et al., 2020; Caron et al., 2021; He et al., 2021; Baevski et al., 2022b). To study E2E fine-tuning of **VC-1**, we use the same policy learning methods described in Appendix C, except we allow the gradients to flow through the **VC-1** backbone and update the weights.

In Table 4, we find an interesting mixed result. In domains that involve large-scale IL or RL (ObjectNav, ImageNav, and Mobile Pick), we use the strategy proposed in Yadav et al. (2023) and observe that adapting **VC-1** with E2E fine-tuning significantly improves performance as compared to using a frozen **VC-1** backbone. Specifically, we see an improvement in ObjectNav success rate (SR) of +7.4 ($60.3 \rightarrow 67.7$), ImageNav SR of +11.3 ($70.3 \rightarrow 81.6$), and Mobile Pick SR of +10.8 ($63.2 \rightarrow 74.0$). Overall, these results suggest that E2E fine-tuning of **VC-1** can achieve the benefits of both task-specialization and domain adaptation. Additional qualitative analysis is provided in Appendix F.

However, in few-shot IL domains (Adroit, MetaWorld, DMC, and Tri-Finger), we find E2E fine-tuning does not result in performance improvement. In fact, in most few-shot IL domains, it leads to a significant drop in performance, a finding that is consistent with prior work (Parisi et al., 2022; Hansen et al., 2022b). We hypothesize that the poor performance of E2E fine-tuning in few-shot IL domains is caused by overfitting, due to fine-tuning a large model with 307M parameters on a small dataset ($\leq 50K$ frames).

**MAE adaptation to mitigate domain-gap.** As an alternative to E2E fine-tuning, we explore adapting **VC-1** with self-supervised learning (SSL). Specifically, in MAE adaptation we continue training the backbone network with the MAE He et al. (2021) pre-training

objective on task-specific data. Then, we freeze these adapted representations and use them to learn task-specific policies. We note that in MAE adaptation, the backbone is adapted using the same data that is used for training the policy (e.g., frames from expert demonstrations), and no additional in-domain datasets are used. While this adaptation strategy cannot address task-specialization, it may serve to mitigate domain gap.

For MAE adaptation, we initialize with **VC-1** weights, and then train with MAE for 100 epochs. In domains where expert demonstrations are available (i.e., Adroit, MetaWorld, DMControl, Tri-Finger, and ObjectNav), we use the RGB frames from these demonstrations for adaptation. In the remaining two benchmarks (ImageNav and Mobile Pick) we sample frames from training environments to create adaptation datasets. Finally, to isolate the importance of initializing with **VC-1** weights, we train in-domain MAE baselines by starting from a random initialization and then following the same approach used for MAE adaptation.

In Table 4, we observe that MAE adaptation substantially improves performance in few-shot learning domains. Specifically, on Adroit performance improves by +18.7 (59.3 → 72.0), MetaWorld by +7.2 (88.8 → 96.0), DMC by +14.0 (66.9 → 80.9), Trifinger by +7.4 (72.7 → 80.1). Interestingly, in DMC and Trifinger, the in-domain MAE baseline (Table 4 row 3) performs surprisingly well, highlighting the importance of in-domain data for representation learning.

Finally, in large-scale IL or RL domains (ObjectNav, ImageNav, and Mobile Pick), we find MAE adaptation results in small reductions in performance from **VC-1** (Table 4 row 4 vs. 6). In these domains, where substantial amounts of data is available for task-specific training (large-scale IL or RL), we find that E2E fine-tuning is the superior approach for adaptation. In aggregate, these results suggests that MAE adaptation should be explored particularly in few-shot domains or when E2E fine-tuning leads to poor performance.

Overall, we find *adapting* **VC-1** results in competitive performance on all benchmarks. On MetaWorld, DMControl, and Tri-Finger **VC-1** with MAE adaptation (Table 4 row 6) is comparable with the best known results (SoTA) and the best results from previous sections (Table 4 rows 1 and 2). Similarly, on ImageNav and Mobile Pick, **VC-1** with E2E fine-tuning (Table 4 row 5) matches or exceeds the best results. Together, these results demonstrate that **adaptation** is a powerful paradigm for using PVRs for EAI.

## 6 DISCUSSION

This work introduced CORTEXBENCH, which comprises of 17 different embodied AI (EAI) task spanning locomotion, indoor navigation, and dexterous and mobile manipulation. Enabled by CORTEXBENCH, we performed the most comprehensive study to-date of visual foundation models for EAI. Specifically, we evaluated state-of-art open-sourced foundation models and find that we do not yet have a strong backbone for all tasks. However, models trained via masked auto-encoders (MAEs) are the most promising. Furthermore, our study finds that naively scaling model size and pre-training data diversity does not improve performance universally across all tasks, but does so on average. Finally, we find that adapting our largest pre-trained model (**VC-1**) results in performance that is competitive with or outperforms the best known results on all benchmarks in CORTEXBENCH.

One of our primary contentions is that in order for the research community to make progress on foundation models for EAI, we need to develop strong benchmarks – for a PVR to be foundational, it must be broadly applicable. Furthermore, as a community we should converge on best practices and a rigorous reproducible experimental methodology; we hope CORTEXBENCH will help the community make progress towards that.

## ACKNOWLEDGEMENTS

The Georgia Tech effort was supported in part by NSF, ONR YIP, and ARO PECASE. The views and conclusions contained herein are those of the authors and should not be interpreted as necessarily representing the official policies or endorsements, either expressed or implied, of the U.S. Government, or any sponsor.

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

Table 5: CORTEXBENCH includes tasks from 7 diverse benchmarks with different combinations of observations, actions, and goals as well as different standard policy learning paradigms.

| Benchmark Suite | Observation Space | Action Space | Goal Specification | Policy Learning |
|---|---|---|---|---|
| Adroit (AD) | RGB + proprio. | Continuous | - | IL |
| Metaworld (MW) | RGB + proprio. | Continuous | - | IL |
| DMControl (DMC) | RGB + proprio. | Continuous | - | IL |
| Trifinger (TF) | RGB + proprio. | Continuous | Goal Image/Position | IL |
| ObjectNav (ON) | RGB + proprio. | Discrete | Object Category | IL |
| ImageNav (IN) | RGB | Discrete | Goal Image | RL |
| MobilePick (MP) | RGB + proprio. | Continuous | Goal Position | RL |

## A    LIMITATIONS

The study presents a thorough examination of visual foundation models but has several limitations. Firstly, in proposing the benchmark, we sought to find a balance between task diversity and the computational resources required for evaluation. However, new and challenging benchmarks in embodied AI, such as those presented in Deitke et al. (2022a), continue to emerge and may merit inclusion in future studies to track progress in this field. Additionally, while we have focused on masked auto-encoders as the pre-training objective and ViT as the architecture in our study, there may be other SSL algorithms that exhibit different scaling behaviors or superior performance on the proposed datasets in our benchmark. Lastly, the adaptation step of the PVR model necessitates separate training on in-domain datasets, as well as careful tuning of hyperparameters such as the number of training epochs and sampling ratio of the dataset. This results in a significant effort to produce a separate adapted PVR model for each benchmark evaluated on our benchmark, and the overall effort increases proportionately with the number of benchmarks included in the study.

In conclusion, it is important to note that although we utilize real-world images and videos for pre-training our visual representation models (PVRs), the evaluation benchmarks used in this study serve as proxies for actual robotic tasks, and thus, the performance of the PVR models on real robots may differ from the rankings established in this study. Further research is necessary to fully evaluate the effectiveness of these models in real-world scenarios.

## B    EMBODIED AI TASKS IN CORTEXBENCH

CORTEXBENCH includes 7 benchmarks (Table 5), illustrated in Figure 1, and described here:

**Adroit (AD)** (Rajeswaran et al., 2018) is a suite of challenging dexterous manipulation tasks in which an agent must control a 28-DoF anthropomorphic hand to perform a variety of tasks. We study the two hardest tasks from Adroit: `Relocate` and `Reorient-Pen`. In these tasks, an agent must manipulate an object into a goal position and orientation, where the goal must be inferred from the scene.

**MetaWorld (MW)** (Yu et al., 2020) is a collection of tasks in which an agent commands a Sawyer robot arm to manipulate objects in a tabletop environment. We consider five tasks from MetaWorld: `Assembly`, `Bin-Picking`, `Button-Press`, `Drawer-Open`, and `Hammer`, which follows the evaluations performed in (Nair et al., 2022).

**DeepMind Control (DMC)** (Tassa et al., 2018) is a benchmark for image-based continuous control in which an agent performs low-level locomotion and object manipulation tasks. We consider five tasks from DMC: `Finger-Spin`, `Reacher-Hard`, `Cheetah-Run`, `Walker-Stand`, and `Walker-Walk`, which follows the work in (Parisi et al., 2022).

**TriFinger (TF)** is a robot, introduced in (Wüthrich et al., 2020), that is composed of a three-finger hand with 3-DoF per finger. We consider two TriFinger tasks: `Reach-Cube` and `Push-Cube`. The `Push-Cube` task was part of the Real Robot Challenge 2020 Real Robot Challenge 2020. We also consider the easier `Reach-Cube` task, which Dittadi et al. (2021) also studies. In these tasks, the agent must either touch the cube with one finger (`Reach-Cube`) or push the cube and move it to a goal location (`Push-Cube`).

**Habitat** (Savva et al., 2019a) is a simulation platform that includes several visual navigation tasks in which agents explore highly photo-realistic unseen 3D environments. We consider two semantic navigation tasks in Habitat: image-goal navigation (`ImageNav`) Zhu et al. (2017) and object-goal navigation (`ObjectNav`) Batra et al. (2020). In both tasks, the agent starts at a random location in an unknown 3D environment and must find a goal location – specified with an image taken from the goal location in `ImageNav` or with the name of an object (e.g., 'chair') in `ObjectNav`. Evaluation is conducted on unseen environments, thus testing the generalization capabilities of the visual encoder and policy.

**Habitat 2.0** (Szot et al., 2021) includes a set of mobile manipulation tasks in which an agent controls a Fetch robot with a 7-DoF arm, mobile base Gu et al. (2022), and suction gripper to rearrange objects in apartment scenes. We consider a challenging version of the `Mobile-Pick` (MP) task from Habitat 2.0, in which an agent must pick up a target object from a cluttered receptacle (e.g., a counter) while starting from a position in which the object is outside of the robot's reach (thus, requiring navigation). We relax the dense goal specification as described in Appendix G.

## C  Downstream Policy Learning

Given a frozen PVR, an agent needs to learn a policy for each task. The EAI community has developed a range of policy learning algorithms from few-shot imitation learning (IL) to large-scale reinforcement learning (RL). For each task in CortexBench, we conform to the community standard for achieving state-of-art performance in that domain.

**"MuJoCo Tasks"** On the tasks from the Adroit, MetaWorld, and DMC suites we train policies using behavior cloning on a small number of expert demonstrations (100 for Adroit and DMC and 25 for MetaWorld), which follows Parisi et al. (2022); Nair et al. (2022). Specifically, we train policies for 100 epochs and report the average rollout performance on the test set for the best intermediate policy during training. For all tasks, the policy is a 3-layer MLP. When using vision transformers (ViT) based PVRs, we use the `[CLS]` token as input to the policy, and with ResNets we use features from the final convolutional layer after global average pooling. These design choices follow prior work such as Radosavovic et al. (2022); Nair et al. (2022).

**"Trifinger Tasks"** For TriFinger, we train policies using behavior cloning on 100 demonstrations per task. Specifically, we train a policy network composed of a 3-layer MLP for 100 epochs for `Reach-Cube` and 1,000 epochs for `Move-Cube`. We report the average score for the best checkpoint over the course of training. As in the "MuJoCo Tasks", the input to the policy is the `[CLS]` token for ViT-based PVRs and average pooled features from the last convolutional layer for ResNet-based models.

**"Habitat Tasks"** We train `ObjectNav` policies with behavior cloning on 77k human demonstrations Yadav et al. (2022c) collected by Habitat-Web Ramrakhya et al. (2022b), totaling 360M environment steps. For `ImageNav` and the Habitat 2.0 `Mobile-Pick` task, we use RL for 500M environment steps with DD-PPO Wijmans et al. (2020) and VER Wijmans et al. (2022). We use patch representations for ViT-based PVRs and grid-features from last convolutional layer for ResNet models, passed through a compression layer Savva et al. (2019a) for a lower dimensional representation for use by the policy layers, which is a 2-layer LSTM for navigation and a 2-layer GRU for manipulation.

## D  Scaling Hypothesis Datasets

### D.1  OpenHouse24 description

The OpenHouse24 dataset (OH24) is a collection of video walk-throughs of furnished residential real estate properties. Over 1600 homes are represented in the dataset, totaling 139 hours of video footage. Each home is traversed in a continuous shot with a stable HD RGB camera by an operator that efficiently visits each room. The dataset represents a diverse set of properties, including (but not limited to) small and large suburban homes, high-rise apartments, ranch homes, and condos. The ensuing walk-throughs range from

| Name | Contains | Total Frames | Frames used |
|---|---|---|---|
| Ego4D | Ego4D | 418,578,043 | 2,790,520 |
| **Ego4D+M** (Manipulation) | Ego4D | 418,578,043 | 2,790,520 |
| | 100DOH | 99,899 | 99,899 |
| | SS-v2 | 25,209,271 | 315,115 |
| | Epic Kitchens | 19,965,439 | 332,757 |
| | Total | | 3,538,291 |
| **Ego4D+O** (OpenHouse24) | Ego4D | 418,578,043 | 2,790,520 |
| | OpenHouse24 | 27,806,971 | 499,442 |
| | Total | | 3,289,962 |
| **Ego4D+N** (Navigation) | Ego4D | 418,578,043 | 2,790,520 |
| | OpenHouse24 | 27,806,971 | 499,442 |
| | RealEstate10K | 10,000,000 | 303,087 |
| | Total | | 3,289,962 |
| **Ego4D+MN** (Manipulation, Navigation) | Ego4D+M | 3,538,291 | 3,538,291 |
| | OpenHouse24 | 27,806,971 | 499,442 |
| | RealEstate10K | 10,000,000 | 303,087 |
| | Total | | 4,340,820 |
| **Ego4D+MNI** (Manipulation, Navigation, ImageNet) | Ego4D+MN | 4,340,820 | 4,340,820 |
| | ImageNet | 1,281,167 | 1,281,167 |
| | Total | | 5,621,987 |

Table 6: Overview of the assembled datasets used for our scaling hypothesis experiments, using up to 5.6M frames.

under a minute to 14 minutes in length, with the average taking 5 minutes and 12 seconds. The dataset will be open-sourced by a separate research project.

# E  Do we already have a foundation model? Additional Plots

We study the distribution of ranks for both the models of scaling hypothesis and the existing PVRs.

In relation to the models we examined for dataset and model scaling, as illustrated in Figure 6, we provide additional evidence of the significance of data diversity. For instance, we observed that the ViT-L models trained in **Ego4D+M** and **Ego4D+N** datasets, achieve the best result in one of the benchmarks, but performs the worst and second-worst in other benchmarks. However, by including data diversity in the **Ego4D+MN** and **Ego4D+MNI** models, we noticed a decrease in the variance of the rank distribution. Notably, the **Ego4D+MNI** model exhibited consistently good performance across all benchmarks and ranks among the top models.

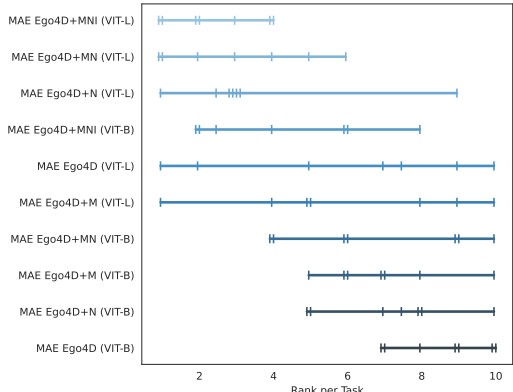

Figure 6: Rank distribution per model - scaling hypothesis.

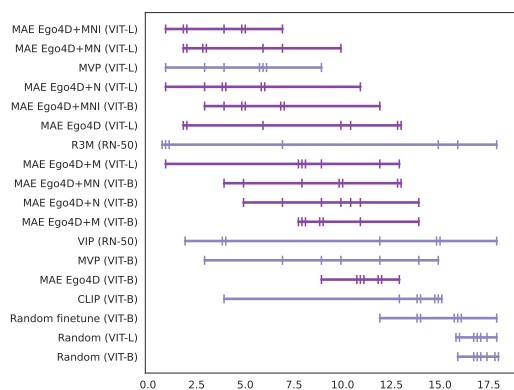

Figure 7: Rank distribution per model - existing PVRs and scaling hypothesis models

## F Attention Visualizations of VC-1

To visualize the attention we apply a mean pooling operation to the attention matrices of the ViT encoder's final layer during inference for downstream tasks. The resulting values are then overlaid onto the image.

We start by noticing the effect of MAE pre-training; frozen VC-1 attention maps appear to focus on the contours and general features of the image. We hypothesize that this results from the MAE reconstruction-based training objective, as contours provide essential information for reconstructing images.

Additionally, we study the attention maps after end-to-end fine-tuning of VC-1 on the downstream tasks. The attention appears to focus on regions of the image that are important for the task (e.g., the objects being manipulated). Thus, through adaptation (via E2E fine-tuning), the model learns to drop attention on areas irrelevant to the specific task.

## G CortexBench Tasks and Training Details

We discuss in more details task specification from Section B in this section.

**ImageNav Benchmark.** Our study conducts `ImageNav` experiments using the standard dataset presented in Mezghani et al. (2021). This benchmark utilizes the Habitat simulator Savva et al. (2019b); Szot et al. (2021) and is situated within the Gibson Xia et al. (2018) environments, which comprise 72 training scenes and 14 validation scenes. The validation set includes 300 episodes for each scene, for a total of 4,200 episodes. In this benchmark, agents are modeled as cylinders with a height of 1.5m, radius of 0.1m, and sensors located 1.25m above the center of the base. The RGB camera has a resolution of 128×128 and a 90° field-of-view. Agent is able to take up to 1000 steps within the environment and are deemed successful if they reach a location within 1m of the goal position and call STOPACTION.

To train the agents within the Gibson environments, we utilize 500M timesteps (25k updates) with 320 environments running in parallel. Each environment collects up to 64 frames of experience, which is followed by 2 PPO epochs utilizing 2 mini-batches. Unless otherwise specified, we use a learning rate of $2.5 \times 10^{-4}$ for training the agents and update the parameters using the AdamW optimizer with a weight decay of $10^{-6}$. We train agents with the reward functions presented in Al-Halah et al. (2022) utilizing the following settings: success weighting $c_s = 5.0$, angle success weighting $c_a = 5.0$, goal radius $r_g = 1.0$, angle threshold $\theta_g = 25°$, and slack penalty $\gamma = 0.01$. We evaluate performance every 25M steps of training and report metrics based on the highest success rate (SR) achieved on the validation set.

**ObjectNav Benchmark.** We present an evaluation of object navigation (`ObjectNav`) using the HM3D-SEM dataset Yadav et al. (2022c). The dataset is comprised of 80 training, 20

**Random ViT-L**        **VC-1**        **Adapted VC-1**

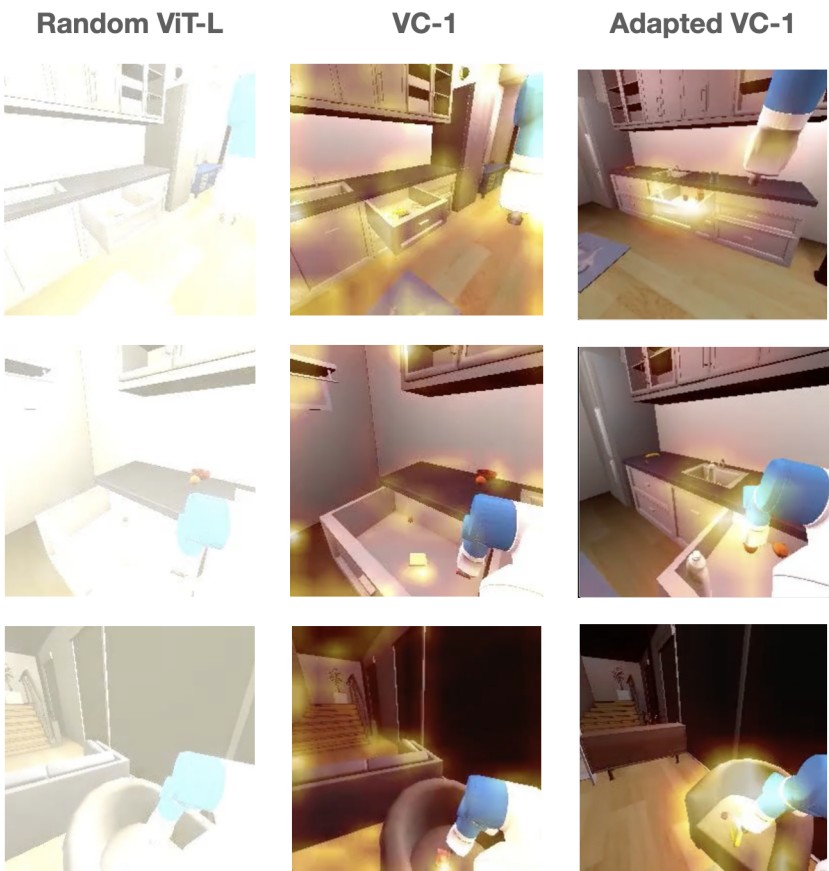

Figure 8: Attention Visualization: We overlay the mean attention matrix in the last layer of the ViT encoder in one of our tasks -MobilePick-. We notice the effect of MAE pre-training on VC-1: The attention focuses in general features of the image; and of task-adaptation: the attention concentrates in task-specific regions of the image

validation, and 20 testing scenes and utilizes the Habitat simulator Savva et al. (2019b); Szot et al. (2021) and HM3D Ramakrishnan et al. (2021) environments. Our results are reported on the v0.1 HM3D-SEM VAL split, which was used in the 2022 Habitat Challenge Yadav et al. (2022a) `ObjectNav` benchmark. The agent in this evaluation is modeled after the LocoBot Gupta et al. (2018) with a height of 0.88m, radius of 0.18m, and sensors placed at the top of the agent's head. The RGB camera has a 640×480 resolution and a 79° horizontal field of view. The task for the agent is to locate objects from one of 6 categories: *'chair'*, *'bed'*, *'plant'*, *'toilet'*, *'tv/monitor'*, and *'sofa'* within 500 steps. Successful episodes are determined by the agent stopping within 0.1m of a viewpoint that is (a) within 1m of any instance of the target object and (b) from which the object is visible, as outlined in the evaluation protocol of Batra et al. (2020).

We utilize a dataset of human demonstrations for training our imitation learning agent in the task of `ObjectNav`. The dataset was collected using Habitat-Web Ramrakhya et al. (2022a); Yadav et al. (2022c) and Amazon Mechanical Turk, and consists of $77k$ demonstrations for 80 scenes from the HM3D-SEM dataset Yadav et al. (2022a). Each scene contains approximately 158 episodes, each with a unique goal object category and a randomly set start location, resulting in approximately 950 demonstrations per scene. The dataset includes a total of ~12.1 million steps of experience, with an average of ~159 steps per episode. By leveraging this human demonstration data, our imitation learning agent is able to learn a more effective policy for navigating to objects in complex environments.

We trained object navigation (`ObjectNav`) agent in the HM3D environment for an approximate total of 400 million steps, utilizing 25,000 updates and 512 parallel environments. Similar to our previous image-based navigation (`ImageNav`) experiments, we employed a weight decay of $10^{-6}$ and utilized different learning rates for the visual encoder and other elements of the model. Specifically, we used a learning rate of $10^{-4}$ for the visual encoder and $10^{-3}$ for all other elements, with the AdamW optimizer. To ensure the quality of our trained models, we evaluated checkpoints after every $10M$ steps and only reported metrics for the checkpoints with the highest validation success rate.

**Habitat 2.0 Rearrangement** We investigate the Habitat 2.0 Rearrangement task proposed by Szot et al. (2021). This task involves a mobile manipulation scenario in which a Fetch robot navigates an ReplicaCAD apartment to pick up a target object from a cluttered receptacle using a mobile base Gu et al. (2022). The robot starts from a non-trivial position and must utilize a variety of sensors, including an egocentric RGB camera, proprioceptive joint sensing, and an object grasping indicator. The action space for the robot includes continuous control over the robot's 7-DOF arm, base movement, and suction gripper. We relax the dense goal specification, where the relative position between the end-effector and the target object must be updated at each step, to a sparse goal specification, where this information is only provided at the start of the episode. This relaxation places greater emphasis on visual input and makes the task significantly more challenging.

**TriFinger Tasks** The TriFinger tasks are implemented in Pybullet. For `Reach-Cube`, the state for the BC policy is $[x_t^{ft}, z_t]$, where $x_t^{ft}$ is the current fingertip position and $z_t$ is the latent visual state vector, obtained by passing the current image observation through the PVR. The success metric captures how close the fingertip is to the optimal distance from the center of the cube, accounting for the half=width of the cube. For `Move-Cube`, the state for the BC policy is $[x_t^{ft}, z_t, \Delta x_g^c]$, where $\Delta x_g^c$ is the goal position for the cube, specified as a displacement from its initial position. Here the success is the distance of the center of the cube to the target goal position. We train a policy network with hidden layers of size 2000 and learning rate $10^{-4}$ for up to 100 epochs for the reach task and 1000 epochs for the move cube task.

## H    Experiment Details of Training PVRs

To train the MAE models, we use the official codebase released by the authors on GitHub (He et al., 2021) and use the default hyperparameters provided by the repo to train the ViT-B and ViT-L models. We found the default values worked well on the CortexBench. However, we do vary the number of epochs we use to train the different models in  Section 4 given the different dataset sizes. We choose the number of epochs per run such that the number of model updates remain constant across all runs and match the number of model updates taken by MAE on the ImageNet dataset. We provide details about the dataset sizes and the epochs calculated for the different runs in  Table 7.

| Dataset Name | Epochs | Frames used |
|---|---|---|
| **Ego4D+N** (VIT-B) | 289 | 3,538,291 |
| **Ego4D+N** (VIT-L) | 289 | 3,538,291 |
| **Ego4D+M** (VIT-B) | 414 | 3,289,962 |
| **Ego4D+M** (VIT-L)) | 414 | 3,289,962 |
| **Ego4D+MN** (VIT-B) | 236 | 4,340,820 |
| **Ego4D+MN** (VIT-L) | 236 | 4,340,820 |
| **Ego4D+MNI** (VIT-B) | 182 | 5,621,987 |
| **VC-1** (**Ego4D+MNI** (VIT-L)) | 182 | 5,621,987 |

Table 7: Experiment Details of Training PVRs.

Table 8: The success rate for each task and each model we evaluate during the study before being aggregated by benchmark.

| task
model | assembly | bin_picking | button_press | cheetah_run | drawer_open | finger_spin | hammer | imagenav | mobile_pick | move_cube | objectnav | pen | reach_cube | reacher | relocate | walker_stand | walker_walk |
|---|---|---|---|---|---|---|---|---|---|---|---|---|---|---|---|---|---|
| CLIP (VIT-B) | 70.7 | 68.0 | 48.0 | 22.7 | 100.0 | 74.6 | 90.7 | 52.2 | 49.8 | 40.1 | 56.6 | 72.0 | 83.8 | 89.9 | 22.7 | 64.9 | 25.4 |
| MAE Ego4D (VIT-B) | 81.3 | 76.0 | 80.0 | 29.1 | 100.0 | 76.9 | 93.3 | 64.0 | 57.4 | 54.0 | 46.8 | 74.7 | 82.6 | 79.8 | 22.7 | 84.3 | 50.5 |
| MAE Ego4D (VIT-L) | 98.0 | 84.0 | 84.0 | 20.7 | 100.0 | 76.5 | 98.7 | 55.8 | 67.6 | 57.0 | 47.6 | 76.0 | 82.4 | 71.9 | 24.0 | 78.4 | 56.3 |
| MAE Ego4D+M (VIT-B) | 76.0 | 58.7 | 84.0 | 31.9 | 100.0 | 75.5 | 98.7 | 65.8 | 59.8 | 57.5 | 47.4 | 77.3 | 80.7 | 89.3 | 25.3 | 80.7 | 44.3 |
| MAE Ego4D+M (VIT-L) | 89.3 | 73.3 | 84.0 | 33.5 | 100.0 | 75.6 | 94.7 | 65.5 | 68.6 | 47.2 | 47.3 | 74.7 | 82.1 | 85.8 | 29.3 | 76.2 | 52.3 |
| MAE Ego4D+MN (VIT-B) | 82.7 | 74.7 | 77.3 | 32.0 | 100.0 | 77.5 | 92.0 | 68.9 | 58.6 | 62.1 | 52.8 | 73.3 | 78.5 | 85.6 | 24.0 | 84.1 | 41.8 |
| MAE Ego4D+MN (VIT-L) | 93.3 | 70.7 | 74.7 | 38.1 | 100.0 | 77.0 | 94.7 | 69.1 | 61.2 | 62.4 | 58.4 | 78.7 | 82.4 | 91.7 | 26.7 | 83.0 | 58.9 |
| MAE Ego4D+MNI (VIT-B) | 88.0 | 78.7 | 82.7 | 32.3 | 100.0 | 76.0 | 98.7 | 67.9 | 60.6 | 60.6 | 55.4 | 76.0 | 83.9 | 82.6 | 32.0 | 83.5 | 44.7 |
| MAE Ego4D+MNI (VIT-L) | 88.0 | 84.0 | 80.0 | 32.8 | 100.0 | 76.8 | 92.0 | 70.3 | 63.2 | 60.2 | 60.3 | 80.0 | 83.3 | 88.0 | 38.7 | 83.3 | 53.7 |
| MAE Ego4D+N (VIT-B) | 86.7 | 76.0 | 73.3 | 28.1 | 100.0 | 75.8 | 96.0 | 68.7 | 59.4 | 54.1 | 54.7 | 77.3 | 81.6 | 78.7 | 22.7 | 72.4 | 42.6 |
| MAE Ego4D+N (VIT-L) | 89.3 | 73.3 | 89.3 | 33.3 | 100.0 | 76.2 | 93.3 | 70.5 | 65.2 | 52.7 | 57.4 | 76.0 | 81.1 | 88.6 | 32.0 | 83.4 | 50.7 |
| MVP (VIT-B) | 92.0 | 73.3 | 92.0 | 33.9 | 100.0 | 76.9 | 98.7 | 64.7 | 56.0 | 44.3 | 51.2 | 69.3 | 75.0 | 86.3 | 26.7 | 84.7 | 47.9 |
| MVP (VIT-L) | 89.3 | 78.7 | 70.7 | 36.9 | 100.0 | 76.4 | 98.7 | 68.1 | 65.4 | 63.4 | 55.0 | 76.0 | 84.8 | 90.2 | 30.7 | 83.2 | 59.3 |
| R3M (RN-50) | 97.3 | 93.3 | 89.3 | 66.1 | 100.0 | 77.1 | 100.0 | 30.6 | 33.2 | 51.9 | 22.6 | 81.3 | 86.5 | 98.4 | 65.3 | 93.8 | 70.1 |
| Random (VIT-B) | 0.0 | 0.0 | 2.7 | 0.4 | 0.0 | 0.1 | 0.0 | 42.1 | 10.8 | 41.3 | 19.2 | 4.0 | 74.3 | 23.4 | 0.0 | 22.7 | 4.0 |
| Random (VIT-L) | 0.0 | 0.0 | 0.0 | 0.5 | 0.0 | 0.2 | 2.7 | 45.2 | 20.6 | 39.4 | 19.3 | 5.3 | 74.9 | 19.9 | 0.0 | 20.1 | 4.6 |
| Random finetune (VIT-B) | 61.3 | 34.7 | 20.0 | 10.2 | 40.0 | 48.6 | 93.3 | 62.5 | 47.6 | 37.6 | 28.5 | 73.3 | 74.5 | 26.8 | 14.7 | 73.6 | 58.1 |
| VIP (RN-50) | 93.3 | 76.0 | 88.0 | 53.2 | 100.0 | 76.1 | 93.3 | 48.8 | 7.2 | 47.2 | 26.4 | 81.3 | 86.2 | 83.2 | 26.7 | 86.6 | 63.4 |

