# OpenReview forum: "Where are we in the search for an Artificial Visual Cortex for Embodied Intelligence?"
_ICLR.cc/2023/Workshop/RRL — RRL 2023 Spotlight_

### Official Review · Reviewer_Ke9e · 2023-02-20
**Large scale study about pre-trained visual representations. Very good scientific contribution. Well written.**

**Rating:** 5
**Confidence:** 3

**Review:**

Review Summary
--------------

Recommendation: oral or poster

__Relevance for workshop: 10/10__

Investigates the factors on how pre-trained visual representations can be helpful for downstream embodied AI / RL tasks.
Fits very well with the workshop topic.

__Scientific quality: 9/10__

Large-scale empirical study that investigates different conditions (source dataset size, diversity, model size).
Procedures are sound.

__Paper quality: 9/10__

Well written.

(Points:  lowest: 0/10 means, highest: 10/10, >=5 means a recommendation to be included to the workshop)


Paper Summary
-------------

The paper is a large-scale study about pre-trained visual representations (PVRs) for embodied RL tasks.
It investigates the influence of different conditions on the performance of PVRs: the size of source datasets, diversity of source datasets, the scale of trained models, and type of models.
It evaluates the performance of different trained PVRs on several target RL tasks.
Key findings show that larger source datasets, higher diversity, and larger scale of model result generally in a better PVR.
Nonetheless, the paper shows exceptions for specific tasks/conditions.
They also trained and provide the current SOA of PVRs in the embodied AI domain.


Major Points
------------

I didn't find major points in your current paper and I think it could be published as is.
The points here are meant for potential future research or to enrich the paper.

1) You currently analyzing the influence of the source datasets on the resulting downstream performance of the PVRs.
It would be interesting to analyze the PVRs themself, such as: how they represent certain features or how the representation change depending on the different source dataset conditions.
And how might different representation types aid the downstream tasks.
This could help to understand why you found exceptions where larger, more diverse source datasets do not improve performance over smaller, less diverse datasets.

2) Your goal is to study the influence of different conditions, such as dataset size and dataset diversity on the trained PVRs.
You do this by using "real" datasets and target tasks where conditions such as diversity are difficult to measure and to control.
You might consider constructing an "artificial" dataset where you have more control over diversity and how well the data fits the downstream task.
Then you could directly manipulate these factors and see their effect on the downstream performance.

3) Influence of training time.
It would be great to see the influence of the training time on the PVRs.
The current paper shows their performance after a set amount of training epochs.
It would be interesting to see how well PVRs do that are trained after a different amount of epochs.
Maybe also, if it is necessary to train them to "completion" on their source dataset.


Minor Points
------------
1) I can not find how many (seeds/repetitions) you ran per downstream task to compute their mean performance and the std.
2) Tables 2 and 4: The light grey text (line 1) is a bit too light. Could be darker.
3) Sometimes you write "ViT" and other times "VIT", but I guess you mean the same thing?
4) Chapter 5.1: All your datasets include the "Ego4D", thus you could avoid this name in your abbreviations to make them look less complex.
   You could use "Man+Nav" to make the M and N easier to recognize for what they stand for.